physiology, cellular biology

social insects, termites, longevity, telomerase, telomeres, kings and queens

**Authors for correspondence:**
Radmila Čapková Frydrychová
e-mail: radmila.frydrychova@hotmail.com
Robert Hanus
e-mail: robert@uochb.cas.cz

†These authors contributed equally to this study.

# Long-lived termite kings and queens activate telomerase in somatic organs

Justina Koubová[1,2,†], Marie Pangrácová[3,4,†], Marek Jankásek[4], Ondřej Lukšan[3], Tomáš Jehlík[1,2], Jana Brabcová[3], Pavel Jedlička[3,5], Jan Křivánek[3], Radmila Čapková Frydrychová[1,2] and Robert Hanus[3]

[1]Biology Centre of the Czech Academy of Sciences, Institute of Entomology, České Budějovice, Czech Republic
[2]Faculty of Science, University of South Bohemia, České Budějovice, Czech Republic
[3]Institute of Organic Chemistry and Biochemistry of the Czech Academy of Sciences, Prague, Czech Republic
[4]Faculty of Science, Charles University in Prague, Prague, Czech Republic
[5]Institute of Biophysics of the Czech Academy of Sciences, Brno, Czech Republic

 JKo, 0000-0002-4097-6097; MJ, 0000-0002-1467-8790; OL, 0000-0002-3201-7486;
PJ, 0000-0001-9324-5151; JKř, 0000-0002-5229-6902; RČF, 0000-0003-2372-4848;
RH, 0000-0002-7054-1975

Kings and queens of termites, like queens of other advanced eusocial insects, are endowed with admirable longevity, which dramatically exceeds the life expectancies of their non-reproducing nest-mates and related solitary insects. In the quest to find the mechanisms underlying the longevity of termite reproductives, we focused on somatic maintenance mediated by telomerase. This ribonucleoprotein is well established for pro-longevity functions in vertebrates, thanks primarily to its ability of telomere extension. However, its participation in lifespan regulation of insects, including the eusocial taxa, remains understudied. Here, we report a conspicuous increase of telomerase abundance and catalytic activity in the somatic organs of primary and secondary reproductives of the termite *Prorhinotermes simplex* and confirm a similar pattern in two other termite species. These observations stand in contrast with the telomerase downregulation characteristic for most adult somatic tissues in vertebrates and also in solitary insects and non-reproducing castes of termites. At the same time, we did not observe caste-specific differences in telomere lengths that might explain the differential longevity of termite castes. We conclude that although the telomerase activation in termite reproductives is in line with the broadly assumed association between telomerase and longevity, its direct phenotypic impact remains to be elucidated.

## 1. Introduction

The reproductive castes of advanced social insects are often very long-lived when compared with their non-reproducing nest-mates and solitary insect relatives. Ant queens and termite kings and queens can live for up to two decades or more, which makes them record holders among insects [1–4]. Evolutionary theories explain the apparent contradiction between the extraordinary lifespan and the high lifelong fecundity of the reproductives by two aspects inherent to eusocial colonies. First, thanks to the life in defendable nests and care provided by working castes, the reproductives are liberated from extrinsic mortality. Second, their fecundity increases in time along with the growing colony population. Both of these factors may select for a longer lifespan [1,3,5].

The debate on proximate mechanisms allowing longevity accompanied by high fecundity has mainly focused on eusocial Hymenoptera [6–8], and only recently have the first studies been published on termites. Elsner *et al.* [9] reported that kings and queens of *Macrotermes bellicosus* upregulate the PIWI-interacting RNA pathway genes, which in turn reduces the activity of transposable elements, known to damage genome integrity [9]. *Reticulitermes speratus* kings upregulate the orthologue of the human breast cancer type 1 susceptibility protein, a

well-known DNA repair agent [10]. Queens of the same species show an increased oxidative stress resistance thanks to the elevated expression and/or activity of catalase, peroxiredoxin and Cu/Zn superoxide dismutase [11,12]. These studies indicate that the longevity of termite reproductives is owing to the recruitment of multiple mechanisms covering the multiplicity of intrinsic causes of mortality.

In this study, we address yet another mechanism of somatic maintenance, mediated by the telomerase, a virtually ubiquitous tool by which eukaryotic cells compensate for age and stress-related shortening of chromosome ends. By adding oligonucleotide repeats to chromosome termini, the telomerase determines the number of potential divisions before the cell reaches senescence [13]. Numerous experiments in vertebrates have provided evidence for the participation of telomerase in cell lifespan control via regulation of its expression, subcellular localization and activity. The telomerase is highly active in germ, embryonic and stem cells but is downregulated in most adult somatic organs to prevent their uncontrolled growth [14]. Studies in vertebrates [15–19] and a few non-vertebrate models [20,21] have confirmed the link between telomere length maintenance and lifespan also at the organismal level. On the other hand, there are examples highlighting that lifespan regulation might be independent of telomere lengths and telomerase activity (TA) [22,23]. Besides the canonical role in telomere maintenance, additional pro-longevity or regulatory functions were ascribed to telomerase or its catalytic subunit, telomerase reverse transcriptase (TERT). These include anti-apoptotic action in cells subjected to stress, protection of mitochondria, nuclear and mitochondrial DNA against reactive oxygen species, transcription modulation of the WNt/β-catenin pathway and regulation of NF-κB-dependent transcription [24–27].

In contrast with the major attention received by telomere biology in vertebrates, the relationship between telomeres, telomerase and longevity has only seldom been addressed in insects, in part because the principal insect model, the fruit fly (Drosophila melanogaster), does not possess telomerase [28]. In the ant Lasius niger, telomeres are significantly longer in females (queens, workers) than in males, but no differences were observed between workers and queens [29]. We recently studied the TA in the American cockroach, Periplaneta americana, and observed a general pattern similar to that in vertebrates: the activity was high in the gonads, while in somatic organs, it gradually decreased during development to low but detectable values in adults [30]. By contrast, we reported a dramatic somatic activation of telomerase during the development and adult life of honeybee queens (Apis mellifera), which contrasted with low activities in adult drones and workers [31].

The observations in honeybees prompted the present study on the relationship between telomerase and lifespan in termites. In the colonies of our main model, Prorhinotermes simplex (Rhinotermitidae), the work tasks are performed by late-stage larvae (fourth stage and older), called pseudergates (hereafter referred to as workers) [32]. The workers retain a full developmental potential: they can remain workers or differentiate into soldiers, winged primary reproductives (founders of new colonies) or wingless secondary reproductives (neotenics), which replace the primaries [32]. From our observations of laboratory colonies, we estimated the maximum lifespan of Pr. simplex workers and soldiers (from egg to death) to be roughly 4 years, similar to other Rhinotermitidae [4]. When a worker differentiates into a reproductive, its potential lifespan

might be extended by several more years. Our data on survival of primary reproductives indicate a median age of 10 years for kings and 12 years for queens, with the maximum recorded so far being 19 years (see the electronic supplementary material, figure S1). Also most neotenic reproductives survive at least 5 more years after their differentiation from workers.

The working stages with a theoretical future reproductive option, as is the case in Pr. simplex, are expected to invest in somatic maintenance to allow their potential long lifespan in the role of reproductives. This expectation has also been supported with empirical data on the upregulation of some pro-longevity and repair functions in working immatures of the termite Cryptotermes secundus (Kalotermitidae) [33]. At the same time, it is reasonable to predict that additional maintenance mechanisms are recruited once a worker obtains the rare opportunity to reproduce and switches its developmental trajectory from the 'dispensable' helper to the long-lived reproductive. In view of this assumption, we study here the telomerase mechanism in termites, with emphasis on the differences between the working stages and the reproductives during their maturation (first year) and maturity (3–5 years). To this goal, we characterize the gene and protein structure of Pr. simplex TERT (psTERT) from genomic and transcriptomic data, quantify the transcript and protein abundances of psTERT, study the enzymatic activity of telomerase and establish the lengths of telomeres. Using these complementary methods, we try to cover the differences and dynamics in telomerase mechanism activation and telomere lengths in all developmental stages and castes, including both types of reproductives of both sexes and known ages, and in different organs. To provide a more general view, we verify the main findings in two other, phylogenetically distant termite species with different social systems, different developmental patterns and longevities of reproductive and working castes.

## 2. Material and methods

### (a) Termites

We used mature colonies of four species. As the main model for all applied techniques, we used Pr. simplex (Hagen) (Rhinotermitidae), kept in laboratory colonies headed either by one pair of primary reproductives or by multiple male and female neotenics. For comparison of telomerase activities and telomere lengths, we used mature laboratory colonies of a phylogenetically relatively basal species Neotermes cubanus (Snyder) (Kalotermitidae), headed by a pair of primary or neotenic reproductives. As another species for TA measurements, we used field-collected mature colonies of Nasutitermes guayanae (Holmgren) from the modern family Termitidae (higher termites), headed by primary reproductives. Finally, as a comparative species for telomere lengths, we used mature laboratory colonies of a congeneric species of the main model, Prorhinotermes canalifrons, headed by multiple neotenics.

Details on breeding conditions of laboratory colonies, the origin of field colonies, social induction of neotenics in Pr. simplex, sampling and sample preparation are provided in the electronic supplementary material, Methods.

### (b) Characterization of Prorhinotermes simplex telomerase reverse transcriptase gene and protein

psTERT structure and transcript variants were identified through the combination of de novo transcriptome assembly from queen ovaries and worker abdomens, genomic DNA sequencing, and rapid amplification of cDNA ends (RACE) as described in the

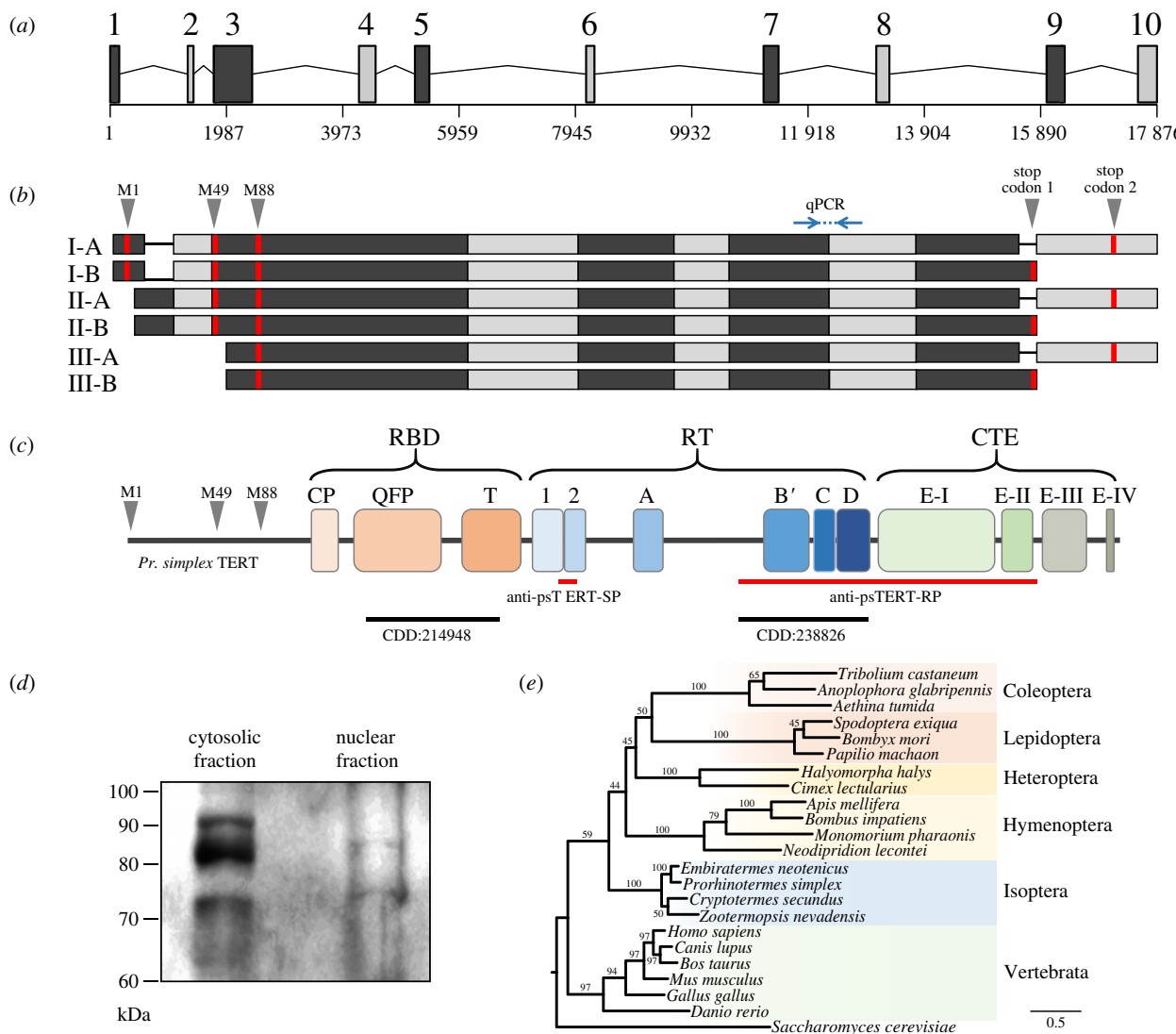

**Figure 1.** Characterization of psTERT. (*a*) Gene structure highlighting 10 exons. (*b*) Transcript isoforms and splice variants. Red bars, initiation and stop codons; blue line, primers and RT-qPCR product. (*c*) psTERT protein structure, labelled according to [34,35]. (*d*) Western blot identifying psTERT isoforms in worker abdomen using anti-psTERT-RP antibody. (*e*) Phylogenetic analysis of psTERT (the longest 809 aa ORF) and TERT protein sequences of other termites, insects, vertebrates and *Saccharomyces cerevisiae* as outgroup. (Online version in colour.)

electronic supplementary material, Methods, along with the primers used (electronic supplementary material, table S1).

## (c) *Prorhinotermes simplex* telomerase reverse transcriptase expression

Isolated RNA was treated with RNase-Free DNase (Promega) and 1 µg was used to synthesize cDNA using SuperScript III (Thermo Fisher Scientific). Real-time quantitative polymerase chain reaction (qPCR) assays were performed using primers targeting the reverse transcriptase domain (figure 1) and with ribosomal protein 49 (RP49) as endogenous control. Primers and reaction conditions are listed in the electronic supplementary material, Methods, table S2. All experiments were run in two technical replicates, averaged for data analysis.

## (d) Telomerase activity and catalytic product

Reverse transcriptase activity of telomerase and identity of synthesized repeats was studied using the Telomeric Repeat Amplification Protocol (TRAP) and its quantitative PCR-based modification (qTRAP).

### (i) Telomerase activity

Proteins were extracted from organs homogenized in 200 µl of extraction buffer (10 mM Tris/HCl, pH 7.6; 1 mM MgCl$_2$;

1 mM EGTA; 0.1 mM benzamidine; 0.1 mM PMSF; 5 mM 2-mercaptoethanol; 0.5% CHAPS; 10% glycerol and 40 U ml$^{-1}$ RNasin Ribonuclease Inhibitors, Promega), incubated for 30 min on ice and centrifuged (12 000 g, 20 min at 4°C). Extracts were collected in supernatants, quantified using the Pierce BCA assay kit (Thermo Fisher Scientific) and stored at −80°C. The qTRAP assay was performed as described in [30], and details are provided in the electronic supplementary material, Methods.

### (ii) Telomerase product identification

The TRAP amplification products were purified with NucleoSpin Gel and PCR Clean-up kit (Macherey-Nagel) and either end-labelled with [γ-$^{32}$P]dATP using T4 polynucleotide kinase (Thermo Fisher Scientific), resolved in 12% polyacrylamide gels and visualized on a Typhoon PhosphorImager scanner system (Amersham Biosciences) or cloned into pGEM-T easy vector (Promega) and sequenced on ABI PRISM 3.1 (Applied Biosystems) with T7 and SP6 primers.

## (e) Telomere structure and lengths
### (i) Fluorescence *in situ* hybridization

To verify the TTAGG repeats localization at chromosome termini of *Pr. simplex*, we performed the fluorescene *in situ* hybridization (FISH) of metaphase nuclei using a (TTAGG)$_n$-specific

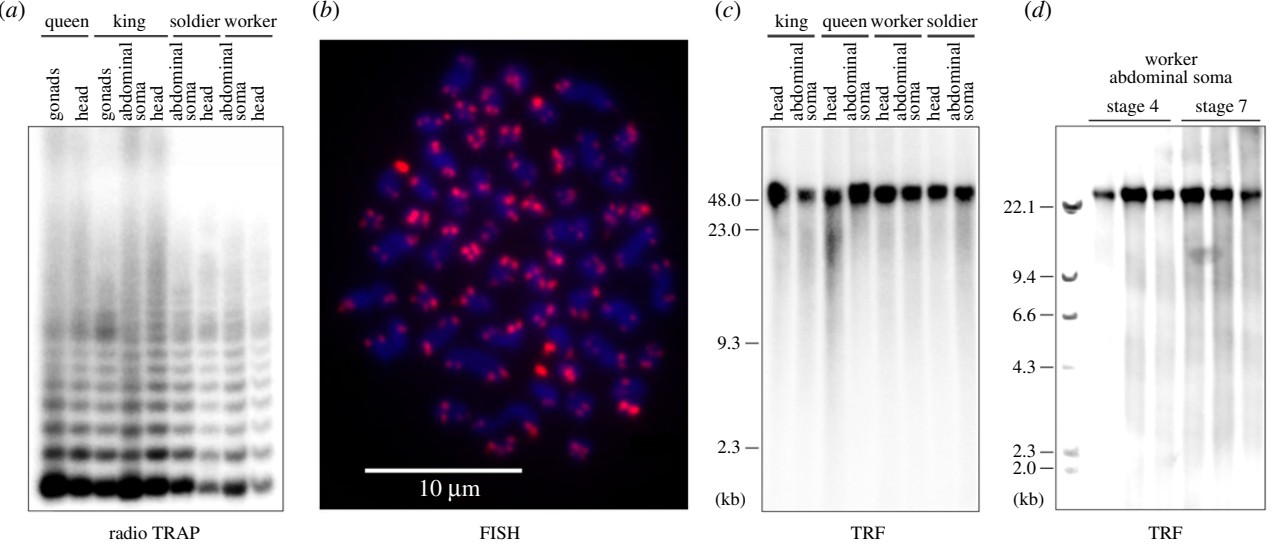

**Figure 2.** Telomerase catalytic activity, identification of telomeric repeats and telomere lengths in *Pr. simplex*. (*a*) Electrophoretic separation of TRAP products. (*b*) FISH of (TTAGG)$_n$-specific probe with metaphase nuclear preparation. (*c*) Terminal restriction fragment (TRF) analysis of somatic genomic DNA (gDNA) from different castes, including 4-year-old neotenic reproductives. (*d*) TRF analysis of somatic gDNA from two different worker stages. (Online version in colour.)

fluorescent probe as described in the electronic supplementary material, Methods.

### (ii) Terminal restriction fragment analysis

High-molecular weight DNA was isolated using phenol : chloroform : isoamyl alcohol extraction. DNA samples (1 μg) were digested with restriction enzymes *RsaI* and *Hinf I* (New England Biolabs), resolved on 0.4–1% agarose gels and stained with ethidium bromide. The DNA was transferred to a positively charged nylon membrane (Hybond-N+; GE Healthcare) by Southern blotting and hybridized to either digoxigenin-labelled probe DIG-(TTAGG)$_4$ or radioisotope-labelled (TTAGG)$_n$ probe. Details are provided in the electronic supplementary material, Methods.

### (f) *Prorhinotermes simplex* telomerase reverse transcriptase quantification

psTERT quantity was evaluated by means of an enzyme-linked immunosorbent assay (ELISA) using de novo developed anti-psTERT antibodies as described in detail in the electronic supplementary material, Methods.

## 3. Results

### (a) Characterization of *Prorhinotermes simplex* telomerase reverse transcriptase gene (*psTERT*) and protein (psTERT)

Using transcriptome assemblies, homology mapping and RACE experiments, we predicted the *psTERT* transcripts and possible splice variants. Reverse mapping on *Pr. simplex* unpublished draft genome revealed the structure of *psTERT* (GenBank accession number MT955237), which spans 18 kb and consists of 10 coding exons (figure 1*a*). We identified three alternative transcription start sites (TSSs) and two alternative end sites (figure 1*b*). The translation in each start isoform can be initiated from its own start codon, all of the three being in-frame (figure 1*b,c*; electronic supplementary material, figure S2). Both end-splicing variants possess proper stop codons followed by poly(A), which altogether indicates six potential open reading frames (ORFs). No internal splicing

with a potential deleterious effect on functional domains was detected, suggesting, together with the substantial length of the ORFs, that the spliced variants may encode functional proteins. Indeed, a high-sensitivity immunodetection using anti-psTERT antibodies indicated several psTERT isoforms in worker soma (figure 1*d*).

A comparison of the longest, 809 aa ORF, with termite and human orthologues and NCBI database of conserved domains revealed high conservation in domains related to canonical TERT function, i.e. the RNA-binding domain (RBD; 63–87% homology with termites, 38% with humans) and reverse transcriptase catalytic subunit (RT; 71–88% termites, 36% humans) (figure 1*c*). The N-terminal extension domain is significantly truncated compared to vertebrates (electronic supplementary material, figure S2). The relative similarity of the psTERT protein sequence with those of other insects and vertebrates is apparent from the phylogenetic tree depicted in figure 1*e*.

### (b) Identification of telomerase products, structure and length of telomeres

We verified the TA in *Pr. simplex* using TRAP assays. In all studied castes and tissues, we detected the characteristic oligonucleotide ladder pattern indicating TA (figure 2*a*). The sequencing of the elongation products confirmed the expected telomeric motif (TTAGG)$_n$.

Using FISH, we localized the TTAGG repeats exclusively at the ends of the 30 pairs of *Pr. simplex* chromosomes (figure 2*b*). Next, we studied the lengths of telomeres by means of terminal restriction fragment (TRF) analysis. In all seven sampled *Pr. simplex* colonies, the terminal fragments of all studied castes and tissues were relatively very long. Their sizes reached up to several dozen of kilobases, with the shortest fragments always being larger than 20 kb, and we did not observe any consistent differences among castes, that might be linked to their differential longevity (figure 2*c*; electronic supplementary material, figure S3). Likewise, when workers of two different stages were compared, i.e. young workers of stage 4 with full developmental options and old workers

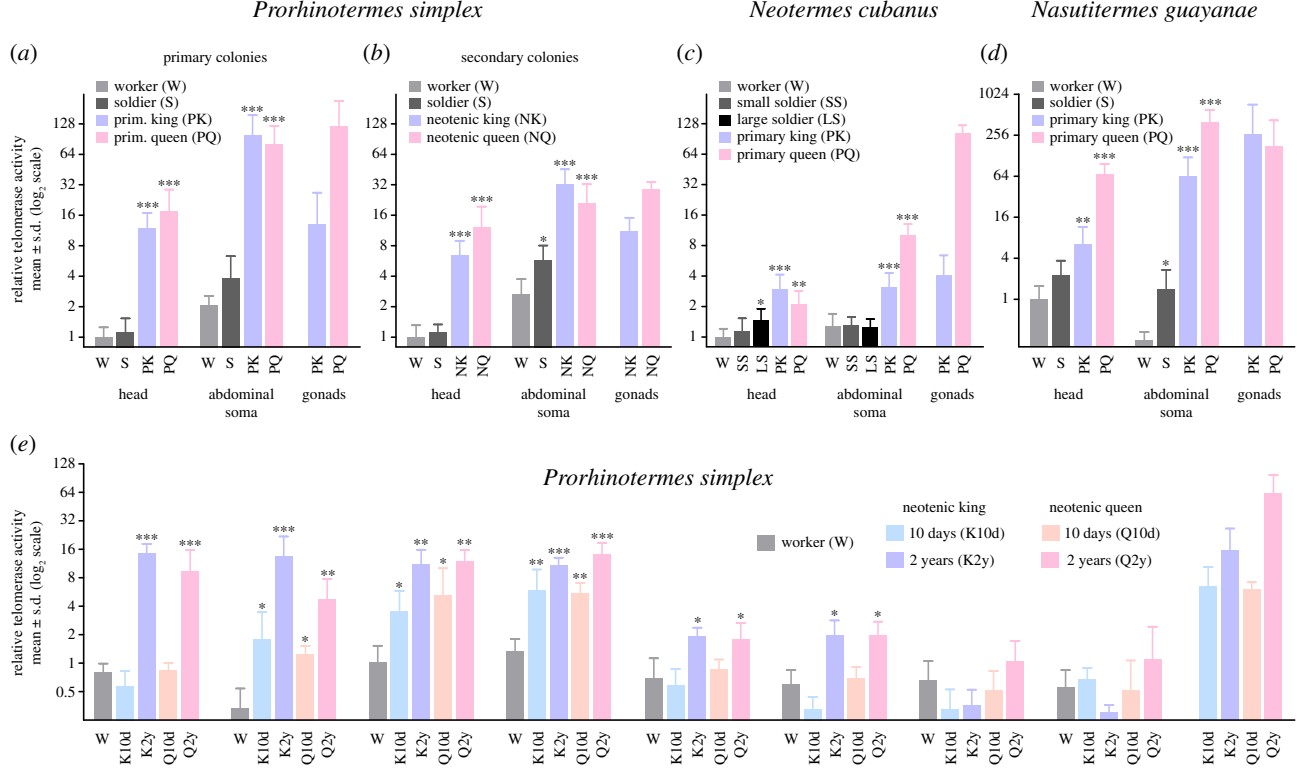

**Figure 3.** TA in somatic and reproductive organs in three termite species. (*a*) TA in three colonies of *Pr. simplex* headed by 3-year-old primary reproductives. (*b*) TA in three *Pr. simplex* colonies headed by 3-year-old neotenic reproductives. (*c*) TA in three colonies of *Neotermes cubanus* headed by 4-year-old primary reproductives. (*d*) TA in three mature colonies of *Nasutitermes guayanae* headed by mature primary reproductives. (*e*) TA in somatic organs of *Pr. simplex* workers and neotenics of two age classes. The data in all graphs are related to the average value in heads of workers from the respective colonies, $\log_2$ transformed and the means compared using ANOVA, followed by the Dunnett's *post hoc* test with worker values for the corresponding organ set as a control (*$p < 0.05$; **$p < 0.01$; ***$p < 0.001$). Detailed test statistics are listed in the electronic supplementary material, table S4. (Online version in colour.)

of stage 7, we observed in both the large telomere sizes with no clear difference between the two stages (figure 2*d*). Analogous results were obtained for two other species, i.e. five colonies of *Pr. canalifrons* and five colonies of *Ne. cubanus* (electronic supplementary material, figure S4).

In parallel, we addressed the identity of *Pr. simplex* telomeric repeats and telomere lengths using the Oxford Nanopore reads of worker genomic DNA. We retrieved multiple reads dominated by $(TTAGG)_n$, many of which were longer than 20 kb (0.3% of all reads over 20 kb), in some cases exceeding 40 kb.

## (c) Telomerase activity in the soma of different castes

The TRAP electrophoretic gels (figure 2*a*) suggested quantitative differences in telomerase products among *Pr. simplex* castes, with the lowest quantities in the soma of workers and soldiers. This initial observation was confirmed in detailed qTRAP analyses, which revealed large differences in TA between working and reproductive castes. In colonies headed by 3-year-old primary reproductives, the average TA was more than 10 and 40 times higher in the heads and abdominal soma, respectively, of kings and queens when compared with workers (ANOVA, $F_{3,12} = 40.2$ for heads and 36.4 for abdomens, $p < 10^{-4}$ for both; Dunnett's test, $q_{(5,3)} > 7.4$ for heads and abdomens in both kings and queens, $p < 10^{-4}$) (figure 3*a*). Likewise, the colonies headed by 3-year-old neotenics showed a highly significant increase of TA in the soma of reproductives ($F_{3,39} = 97.5$ for heads and $F_{3,37} = 73.3$ for abdomens, $p < 10^{-4}$ for both; $q_{13,9} = 10.8$ for heads of kings,

$q_{(13,8)} = 13.8$ for heads of queens, $q_{(12,8)} = 12.8$ and 10.1 for abdomens of kings and queens, respectively, all $p < 10^{-4}$), though the relative differences were less dramatic than in primary colonies (figure 3*b*). In both types of colonies, TA in the gonads was roughly corresponding (ovaries) or inferior (testes) to that in abdominal soma of reproductives. TA in soldiers was slightly, sometimes significantly, higher in comparison with workers. See the electronic supplementary material, table S4 for detailed statistics.

## (d) Telomerase activity in other termite species

In both comparative species, *Ne. cubanus* and *Na. guayanae*, somatic TA upregulation in reproductives was confirmed ($F_{4,27} = 11.7$, $F_{4,26} = 38.5$ for heads and abdomens of *Ne. cubanus*, respectively; $F_{3,24} = 30.2$, $F_{3,16} = 56$ for heads and abdomens of *Na. guayanae*, respectively, all $p < 10^{-4}$) (figure 3*c,d*). The differences in TA between working castes and reproductives were smaller in *Ne. cubanus* when compared with those in *Pr. simplex* and especially *Na. guayanae*, in which the TA in king and queen abdomens and queen heads reached 250-fold ($q_{(7,3)} = 7.9$, $p < 10^{-4}$), 1500-fold ($q_{(7,3)} = 11.6$, $p < 10^{-4}$) and 70-fold ($q_{(11,3)} = 9.26$, $p < 10^{-4}$), respectively, of that recorded in workers. In both species, TA values in soldiers were comparable or slightly higher than those in workers (see the electronic supplementary material, table S4 for details).

## (e) Telomerase activity in individual somatic organs

We deciphered the TA in the somatic organs of *Pr. simplex* workers and neotenics of two age classes, i.e. 10 days and

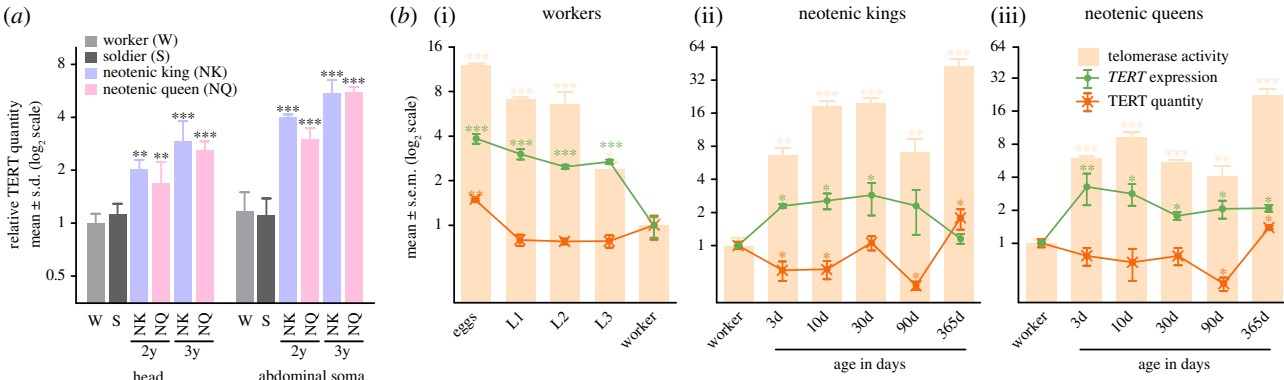

**Figure 4.** Telomerase activity, *TERT* expression and TERT quantity in *Pr. simplex* soma during the development. (*a*) Relative quantity of psTERT in heads and abdominal soma quantified using ELISA. Four pairs of two-year-old (2y) and three pairs of 3-year-old (3y) neotenics were sampled from two colonies together with four workers and soldiers. (*b*) Telomerase activity, *TERT* expression and TERT quantity during the development of workers (i), neotenic kings (ii), and neotenic queens (iii). Three to four replicates per phenotype were prepared, obtained values related to the average from workers from the respective colony, $\log_2$ transformed and the means compared using ANOVA, followed by the Dunnett's *post hoc* test with worker values set as a control (*$p < 0.05$; **$p < 0.01$; ***$p < 0.001$). Detailed test statistics are listed in the electronic supplementary material, table S5. (Online version in colour.)

2 years after their differentiation from workers (figure 3*e*; electronic supplementary material, table S4). In most organs, we observed a significant (mandibular glands, head capsule; ANOVA + Dunnett's test, $p < 0.05$) or highly significant (nerve cord, digestive tube, epidermis, fat body; $p < 10^{-2}$–$10^{-4}$) increase in TA between the worker stage and mature neotenics; in some cases, this pattern was already significant in 10-day-old neotenics.

### (f) *Prorhinotermes simplex* telomerase reverse transcriptase quantity in somatic tissues of workers and reproductive castes

We quantified psTERT abundance using anti-psTERT antibodies and ELISA (figure 4*a*; electronic supplementary material, table S5). psTERT quantity in the soma of mature neotenics was significantly higher than in workers, and the differences were bigger in abdomens than in heads ($F_{5,18} = 16.5$ for heads and $62.4$ for abdomens, $p < 10^{-4}$ for both). An age-dependent increasing trend in psTERT quantity could be observed between 2- and 3-year-old neotenics, reaching a maximum of nearly fivefold in the abdomens of older neotenics of both sexes when compared with workers ($q_{(4,4)} > 11.7$ for both, $p < 10^{-4}$).

### (g) *psTERT* expression, psTERT abundance and telomerase activity during postembryonic development

Next, we followed the dynamics of somatic TA, *psTERT* expression and psTERT abundance during the development of *Pr. simplex* workers and neotenics (figure 4*b*; electronic supplementary material, table S5). We observed a gradual significant decrease of TA and *psTERT* expression from maximum values in embryos (eggs), through three larval stages, to the lowest values in workers ($F_{4,15} = 52.7$ for TA and $17.6$ for TERT quantity, $p < 10^{-4}$ for both). psTERT quantity decreased significantly between the embryo and the first larval stage, and then remained at relatively stable levels ($F_{4,14} = 44.4$, $p < 10^{-4}$).

A significant increase in TA was recorded from the first days after the neotenic differentiation ($F_{5,12} = 26.19$, $p < 10^{-4}$), and the maximum activity was observed in the soma of mature, 1-year-old neotenics (40 and 20 times higher in males and females, respectively, compared to workers; $q_{(3,3)} > 10.2$ and $p < 10^{-4}$ for both). *psTERT* expression roughly followed the trend of TA, however, without the final increase to peak values in 1-year-old reproductives. psTERT quantity showed a fluctuating trend, but consistently with the previous experiments (figure 4*a*), we observed its significant increase in 1-year-old neotenics, compared to workers (see the electronic supplementary material, table S5 for details).

## 4. Discussion

In this study, we tested the possible correlation between the activation of the telomerase mechanism and differential longevities of termite castes. In our main model, *Pr. simplex*, we characterized the gene and protein structure of psTERT, including transcript and protein variants, quantified psTERT transcript and protein abundances, telomerase activities and lengths of telomeres across different castes, life stages and organs, with emphasis on the differences between reproductives and workers, from which the reproductives develop. We demonstrated a strong association between the life expectancies of different castes and the somatic activation of telomerase. In the soma of long-lived primary and neotenic kings and queens of *Pr. simplex*, we observed a conspicuous increase in telomerase catalytic activity and quantity of the psTERT protein when compared with workers. We confirmed the main finding on the somatic telomerase activation also in the reproductives of two other species with different caste systems and developmental patterns. Telomerase activation also took place in tissues, which are *a priori* expected to have low mitotic activity, like the ventral nerve cord. These results contrast with the conventional view of telomerase, whose activity should be low or absent in adult somatic organs with low proliferative potential. In parallel, we studied the lengths of telomeres in *Pr. simplex* and two other termites. The telomeres in all phenotypes and species were relatively long (up to over 50 kb) and did not show length differences that might explain the differential longevity or observed differences in telomerase activities and

*Proc. R. Soc. B* **288**: 20210511

abundances. We conclude that although our observations are in line with the broadly assumed association between telomerase and longevity, the potential pro-longevity mechanism of telomerase in reproductives remains elusive. We discuss below individual findings in the light of this general conclusion.

## (a) *Prorhinotermes simplex* telomerase reverse transcriptase gene and protein structure

psTERT gene and protein structures are highly homologous to those of other termites, insects and vertebrates. The 18 kb *psTERT* gene consists of 10 coding exons, a number comparable with most insect taxa [36]. *TERTs* in the termites *Zootermopsis nevadensis* (NW_019059798.1, LOC110829667) [37] and *Cryptotermes secundus* (NW_019723906.1, LOC111862026) [38] differ in gene sizes (14.3 kb and 28.9 kb, respectively), but share the same exon numbers, junctions and compositions. Most domains essential for TERT reverse transcriptase function are highly conserved. The RBD domain includes three conserved motifs CP, QFP and T, important for TA in yeast [34]. A high conservation is retained in reverse transcriptase motifs A, B′, C and D [39]. Amino acid residues essential for metal cofactor binding in mammals (D712, D868 and D869 in humans), phosphorylation site Y707 associated with telomerase shuttling to cytoplasm under oxidative stress and the hydrophobic residue V867 needed for transcription fidelity in humans are fully conserved [40,41]. A high homology is preserved in the CTE domain, important for the canonical function of human telomerase [35] (electronic supplementary material, figure S2).

Post-transcription regulation by alternative splicing is crucial for human telomerase in its canonical function, as well as for some alternative roles ascribed to TERT [42]. Though widespread in other animals, including insects [36], TERT alternative splicing has only rarely been functionally addressed [43]. In *Pr. simplex*, we identified as many as six possible *psTERT* splice variants. Their size and lack of deleterious splicing indicate multiple functional proteins, as also suggested by immunodetection of several psTERT isoforms. The splicing of the CTE motifs E-III and E-IV (exon 10) may have functional consequences [35], while the splicing of exon 1 may affect the intra-cellular distribution of psTERT. A strong nuclear localization signal RRRKKKIK [44] in the N-terminal region of TSS2 variants suggests the transport of the protein through nuclear pores, while the signal peptide MFRSCLTIFRVRCYRAVVFWVVTLCSLLTDC [45] in TSS1 isoforms may be linked to vesicular transport to other membrane-bound organelles. In conclusion, *psTERT* structure indicates the ability of psTERT to perform its canonical function, but also some features associated with non-canonical functions remained conserved [40]. The presence of multiple splice variants may allow for differential localization and functional divergence of TERT isoforms.

## (b) Identification of telomerase products, structure and length of telomeres

We confirmed that termite telomeres are composed of TTAGG motifs. This ancestral arthropod telomeric motif was previously recorded in termites and most other Polyneoptera, including the close termite relatives, the cockroaches, mantises and stick insects [46]. Mean telomere lengths in most insects range between units to 20 kb [29,47], but in some taxa, they

can reach up to several dozens of kilobases [48,49]. In the three studied termite species, we observed telomeres ranging from more than 20 up to more than 50 kb without conspicuous differences among castes or tissues, or differences related to the absolute age of workers, that would correlate with differential life expectancies. Similar conclusions were also drawn for eusocial Hymenoptera; significant differences in telomere lengths between workers and queens could not be detected neither in the ant *L. niger* [29] nor in the honeybee [31]. All these observations suggest that telomere shortening does not determine the shorter lifespan of social insect workers. We can speculate that the relatively long telomeres in termites, also recorded in other advanced Polyneoptera [49], are a common preadaptation for the relatively long life of these insects in general.

## (c) Telomerase activity

Non-negligible TA was previously reported in the soma of late stages and the adults of several insects [29,30,50–52]. This may signal a less strict regulation of telomerase when compared to vertebrates, in which it is inactive in most adult organs. Nevertheless, a more detailed analysis in the cockroach *Pe. americana* revealed a regulated pattern of somatic TA decreasing gradually from maximum values in embryos through larval stages to basal TA in adults [30]—a scenario, which roughly correlates with observations made in vertebrates and with dynamics in cell proliferation. The situation is different in the long-lived castes of social insects. In a recent study on the European honeybee, we reported that larvae, pupae and adult drones and workers display the expected decrease of TA from high embryonic values, while queen third-stage larvae and adult queens show a marked increase of somatic TA [31]. In spite of fundamental differences in postembryonic development between holometabolous bees and hemimetabolous termites, the general pattern of telomerase activation in long-lived reproductives is common to both. Just as in the honeybee, we documented in *Pr. simplex* a stepwise decline in TA from embryos through larval stages to workers, contrasting with a great increase of TA in the soma of mature reproductives of both sexes, accompanied also by a significant increase of TERT quantity.

Two other termite species showed a similar trend in somatic telomerase activation in reproductives. Interestingly, the relative increase in TA was lower in the reproductives of the phylogenetically basal species *Ne. cubanus*. Working immatures and reproductive castes of Kalotermitidae are relatively long-lived (units of years), with low differences in life expectancies, and the queen body size and fecundity are also low [4]. By contrast, somatic telomerase activation was particularly high in queens of the socially advanced species *Na. guayanae*, endowed, like other higher termites, with high fecundity, large body size and a high metabolic rate. Also the lifespan differences between reproductives and workers are the highest in higher termites (years versus months) [4]. It appears, thus, that the rate of telomerase somatic upregulation in reproductives is correlated with lifespan differences between working and reproductive castes, and with the overall fecundity of the reproductives. The observed interspecific trend also corresponds to the presumed differences in somatic maintenance investments by the working immatures according to their reproductive potential in different species, formulated by Monroy Kuhn

[33]. The relative difference in TA between the working stage and a reproductive is the lowest in *Ne. cubanus*, whose immatures have a high probability to reproduce as primaries or inherit the nest as neotenics, it is higher in *Pr. simplex*, whose large population of immatures provides lower chances to reproduce, and reaches the maximum in *Na. guayanae*, in which the workers are irreversibly sterile.

The conclusions depicted above for lifespan differences and telomerase upregulation in reproductive individuals also hold for the eusocial bees. Unlike in the honeybee, having high lifespan differences and great telomerase upregulation in queens, we observed a low somatic TA in mature post-diapause queens of the bumblebee *Bombus terrestris*, in which the lifespan differences between queens and workers are essentially only owing to queen overwintering [52].

We observed the highest telomerase activation in the ventral nerve cord and fat body of *Pr. simplex* reproductives. Interestingly, the highest relative TA was also recorded in the post-mitotic nerve tissue in honeybee queens, more specifically in the brain [31]. Although adult fat body cells of termite queens (and sometimes also kings) undergo endoreduplications [53], these are expected to be infrequent in *Pr. simplex*, belonging to termites with low fecundity and low adult abdominal growth [54]. In other words, in both the honeybee and termites, the telomerase activation in long-lived reproductives appears as independent of the actual rate of nuclear divisions, at least in some of the studied organs.

## (d) Perspectives

While we are currently unable to establish the real function of telomerase activation in long-lived termite reproductives,

some of the questions arising from our results may be realistic targets for future investigation. Among them, differential expression of the multiple identified splice variants leading to several potential TERT isoforms with different cell localization signals and possibly also functions should be studied in individual life stages and organs with respect to life expectancy and the actual age of the termites. Future studies should consider both the canonical telomerase role in telomere extension as well as the alternative functions ascribed to this enzyme, such as the frequently reported role in oxidative stress reduction. The latter hypothesis is especially appealing, as previous studies have shown an important investment into oxidative stress compensation in termite queens and termites in general [11,12,55].

Data accessibility. psTERT gene and protein sequences including the six predicted variants have been deposited in GenBank under accession no. MT955237. RNASeq reads used for gene structure reconstruction and variant predictions have been deposited in the BioSample database under accession nos. SAMN17050365-72.

Authors' contributions. R.Č.F. and R.H. designed the study; J.Ko., T.J. and R.Č.F. performed TRAP, qTRAP and TRF experiments; M.P. and P.J. performed RT qPCR analyses; M.P. and J.B. quantified psTERT; O.L. studied psTERT structure; J.Kř. prepared samples; M.J. performed FISH imaging; R.H., R.Č.F., M.P. and J.Ko. drafted the manuscript, with contributions from other authors.

Competing interests. We declare that we have no competing interests.

Funding. This research was funded by the Czech Science Foundation (18-21200S), ELIXIR CZ (MEYS) (LM2015047) and Institute of Organic Chemistry and Biochemistry, CAS (RVO: 61388963).

Acknowledgements. We are grateful to the Laboratoire Environnement HYDRECO for logistical support. We acknowledge the assistance of Michala Sábová with TRAP assays, and Jiří Fajkus and Vratislav Peška with Southern hybridization.

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
