## [Peer Review File · Proceedings of the Royal Society B: Biological Sciences]

Review History

RSPB-2020-2439.R0 (Original submission)

Review form: Reviewer 1

Recommendation

Major revision is needed (please make suggestions in comments)

Scientific importance: Is the manuscript an original and important contribution to its field?

Excellent

General interest: Is the paper of sufficient general interest?

Excellent

Quality of the paper: Is the overall quality of the paper suitable?

Excellent

Is the length of the paper justified?

Yes

Should the paper be seen by a specialist statistical reviewer?

No

Do you have any concerns about statistical analyses in this paper? If so, please specify them explicitly in your report.

No

It is a condition of publication that authors make their supporting data, code and materials available - either as supplementary material or hosted in an external repository. Please rate, if applicable, the supporting data on the following criteria.

Is it accessible?

Yes

Is it clear?

No

Is it adequate?

Yes

Do you have any ethical concerns with this paper?

No

Comments to the Author

Major comments

1. You make use of a multitude of different methods, and you make use of multiple species which is great. Emphasize this at the end of the introduction and the beginning of the discussion. At the moment the connection and reasoning of the different approaches is not always clear.
2. Some information in the material and methods section is missing including the statistical analyses (only rudimentary mentioned in the text, some details only in the figure legend).
3. Information on the RNAseq data and transcriptome are not complete including data deposition information.

Minor comments:

Abstract

First sentence: maybe rather start with eusocial insects in general and not directly with termites in particular

Line 28: rephrase "neotenic" reproductives for non-termite experts (i.e. secondary reproductives)

Introduction

Line 41: change "for up to two decades" to e.g. "for several decades". The tables in the Keller 1998 tables indicate that some ant and termite queens manage to live for up to 30 years, and I think there is evidence for >40 in termites.

Line 42: change "admirable lifespan" to either "extraordinary lifespan" or "long lifespan".

Line 47: change "at a time" to "simultaneously".

Line 48: please add some citations for Hymenopteran studies, e.g. from the Foitzik (Temnothorax) and Heinze (Platythyrea, Cardiocondyla) groups.

Line 60: either "the telomerase determines" or "telomerases determine"

Line 63: again, either "The telomerase is" or "Telomerases are"

Line 73: To what extent does "Wnt/ β -catenin pathway" play a role in longevity, anti-oxidation...?

Line 81: remove the sentence "Telomeres and telomerase have rarely been studied in social insects." Which is redundant to the first sentence of this paragraph

Line 83: how about moving the Apis sentence after the cockroach, then you could say "Moreover, we recently..." instead of twice "in our recent study..."

Line 96: add reference to "the recorded maximum was 19 years"

M&M

Line 106 + Line 117: change "are" to "were"

Line 114: how many workers and soldiers?

Line 119: please add coordinates of sampling site. What about sampling permission and Nagoya certificate?

Line 120: change to “were kept in the laboratory”

Line 122: “under a stereomicroscope” and either say “the digestive tube” or “digestive tubes”

Line 126: is “alternative splicings” the correct term? Alternatively “alternative splice variants” or “isoforms”... && add information that you conduct a de novo transcriptome assembly

Line 130: did you run technical replicates?

Line 128 +170+189: “in the electronic”

Line 152: “Pierce BCA assay kit” manufacture information missing

Supplementary Methods

Page2: add information on how DNA was isolated, which protocol did you use, special high molecular weight DNA? Add information on what type of sequencing you are talking about. I assume Nanopore sequencing? How many flow cells?

You extracted DNA from a pool of 8 individuals. Why were “barcoded” libraries loaded, why did you need barcodes for a single library?

“according to 70-day immunizationProtocol”: add reference

“by ELISA” + “in Western blot”: add information

Results

Line 195: “mapping on the newly”

Line 195: do you mean transcriptome instead of genome? Otherwise give genome reference

Line197: does the accession number refer to the psTERT sequence or the transcriptome? Do you make the transcriptome accessible somewhere?

Lines 206-213: The workflow behind these results including the construction of the phylogeny is completely missing from material&methods

Figure 1E = phylogeny: please add the sequence of at least one ant species. Since ants belong to the other eusocial taxa besides termites, it would be highly interesting to them represented in the phylogeny. Please run the ML for at least 1000 replicates.

Line 256: please add the test statistics to the p-value and also add to the M&M section which program and test you used for the statistical analyses.

Discussion

In the beginning of the discussion where you add a short summary of your study add information that you also investigated splice variants, spTERT expression in various tissues and even in multiple species.

Line 286: “of the psTERT protein”

Line 302: “but share exon numbers, junctions and compositions” do you mean “but share the same exon numbers”?

Line 303: “The RBD domain”

Line 314: would it be possible to determine which splice variants are expressed in which tissue, or at least how many of the 6 are expressed in which tissue? If this information could be obtained easily I think it would be an interesting point to address.

Review form: Reviewer 2 (Judith Korb)

Recommendation

Major revision is needed (please make suggestions in comments)

Scientific importance: Is the manuscript an original and important contribution to its field?

Good

General interest: Is the paper of sufficient general interest?

Good

Quality of the paper: Is the overall quality of the paper suitable?

Good

Is the length of the paper justified?

Yes

Should the paper be seen by a specialist statistical reviewer?

No

Do you have any concerns about statistical analyses in this paper? If so, please specify them explicitly in your report.

Yes

It is a condition of publication that authors make their supporting data, code and materials available - either as supplementary material or hosted in an external repository. Please rate, if applicable, the supporting data on the following criteria.

Is it accessible?

N/A

Is it clear?

N/A

Is it adequate?

N/A

Do you have any ethical concerns with this paper?

No

Comments to the Author

The authors studied the potential role of telomere in explaining lifespan differences between queen/king and workers in termites. Social insects are emerging model organisms of ageing as they realize very different lifespan between colony members that share the same genetic background with queens/kings that can have extraordinary longevity for an insect. The authors identified and characterized the telomerase reverse transcriptase (psTERT) in the termite *Prorhinotermes simplex*, quantified psTERT expression, measured telomerase activity and determined telomere lengths and location. They used different caste, tissues and age classes and did some of the experiments also with other termite species. They show that queens and kings have higher telomerase activity, also in most tissues of the soma where it is just as high as in the gonads. But telomere length does not seem to differ between castes.

This is a comprehensive study that addresses a (potentially) important ageing mechanism that received relative low attention in social insects so far and had not been studied in termites. Therefore, it is well-suited for PRSB. The manuscript is also clear and well-written. I have a number of questions/topics that should be addressed in a revision, most of them concerning methods and statistics, but also interpretation of results.

Major points

1) I am wondering a bit about the assumption of the study that workers should invest less in anti-ageing. As mentioned in the manuscript, workers in termites like *P. simplex* are immatures from which the reproductives develop (e.g. L. 88). Hence, evolutionary theory predicts that they should invest in anti-aging mechanisms as they are individuals that have not reached maturity yet – in contrast to species with sterile workers or at least a bifurcated

development (see also Monroy Kuhn et al 2019 PLoS One). Please include this aspect into your considerations and conclusions.

2) Does the lifespan between neotenic and primaries really differ? While for the neotenic a range is given for the primaries only a maximum is provided (1.95-97). As you seem to have the data (1.108/109), please add a life table (e.g. Kaplan Meier) analyses. This would also help in interpretation the results (see below)

3) Methods: Many more details are needed in the methods. The samples sizes for each of the experiments should be provided as well as whether the samples came from the same of different (stock)colonies. If samples came from the same colony (or stock colony) this must be included in the analyses as they are not independent. Please provide more information on the genomic sequencing and the assembly protocol as well as the Oxford nanopore methods. Are these data made available? How many colonies did you use for the *Neotermes cubanus*, *Nasutitermes guayanae* and *Prorhinotermes canalifrons*. Did you have to pool sample to analyse, for instance, specific tissues; how many samples did you pool. Please add for each analyses.

4) Statistics: Please add a statistics section to the methods which mentions how the different data sets were analysed. Also add the tests with the typical test characteristics (N or df, F-value...) to the brackets with the P-values in the text. How did you deal with non-independency of data and did you test for assumptions of the tests. Often such data the error distributions of such data are not normal-distributed. Currently the only information provide, is in the legends which is too limited

5) Discussion: (1) One reason why no differences in telomere lengths was found is that the reproductives/workers were not old enough, i.e. no senescence processes were occurring yet. This is a difficult issue as one need survival curves to determine how organisms age. For social insects, there is, e.g. some evidence that suggests that reproductives 'suddenly' age, i.e. there is no gradual senescence. Its, seems that you can do such analyses given that you seem have kept colonies over longs periods. Please, add them to your data. Also mention the ages of the individuals used for the telomere lengths analysis. (2) (partly linked to 1): Another reason why you seen no differences in telomere length between castes may exactly be the telomerase activity. If they protect the telomeres from shortening, then one would not expect to see differences between castes. This might also lead to non-linear scenescene. This is a bit speculative also given that it is currently unclear to me what the sample sizes for the telomere length estimates are. Yet, I think it would be worthwhile considering.

Minor comments

l. 114-115: unclear. Does *P. simplex* has one pair of reproductives or is it polygamous?

l. 117: change to: ...reproductives of known age...

l.145 and the following: Can you please explain how the qTRAP assay functions? How can one do a PCR with proteins?

l. 215-229: Were these the results of single samples? $N = 1$; as implied from Figure 2? Please state clearly. With a $N = 1$ one cannot really say whether it is higher or lower in one caste/ tissue , or not. Can you also show the data for *N. cubanus*.

l. 237: can you specify: how old were they?

Discussion: The discussion is clearly written. I think it would further benefit the reader to have subheadings that characterize the different topics addressed.

l. 300, 301: Please cite the papers which published the two termite genomes. Please, add the original Gene ID as named in the genomes

l. 353: change to: .. phylogenetically basal..

l. 354: The workers of the *Kalotermitidae* are not sterile (see also Abstract, and adjust). In all species with a linear development, they are totipotent immatures. This is important consequences regarding the evolution of ageing; see above and Monroy Kuhn et al 2019 PLoS One.

Decision letter (RSPB-2020-2439.R0)

30-Nov-2020

Dear Dr Hanus.

Thank you for the submission of your manuscript RSPB-2020-2439 entitled "Long-lived termite kings and queens activate telomerase in somatic organs". We have now received received referees' reports on the manuscript, and these have been evaluated by the Associate Editor.

The manuscript has, in its current form, been rejected for publication in Proceedings B. This action has been taken on the assessment of the Associate Editor and the advice of the referees, who have provided very comprehensive reviews that indicate that substantial revisions are necessary. Nevertheless we are all agreed that this is a potentially valuable and very interesting manuscript, and with this in mind we would be happy to consider a resubmission, provided the comments of the referees are fully addressed. However please note that this is not a provisional acceptance.

Finally, I hope you and your co-authors are well in this challenging year.

Yours sincerely,
Professor Loeske Kruuk
mailto:proceedingsb@royalsociety.org

Associate Editor
Comments to Author:

The authors studied role of telomere length in explaining differences in longevity between reproductives and workers in termites. The paper has been reviewed by two experts in the field, and, while both find some merit in this work they also expressed several important concerns that cover both the methodology and the interpretation of the findings. Therefore, I cannot recommend this paper for publication in its current form. However, the Reviewers made very specific suggestions that can be used in the revision.

Reviewer(s)' Comments to Author:

Referee: 1

Comments to the Author(s)

Major comments

1. You make use of a multitude of different methods, and you make use of multiple species which is great. Emphasize this at the end of the introduction and the beginning of the discussion. At the moment the connection and reasoning of the different approaches is not always clear.
2. Some information in the material and methods section is missing including the statistical analyses (only rudimentary mentioned in the text, some details only in the figure legend).
3. Information on the RNAseq data and transcriptome are not complete including data deposition information.

Minor comments:

Abstract

First sentence: maybe rather start with eusocial insects in general and not directly with termites in particular

Line 28: rephrase "neotenic" reproductives for non-termite experts (i.e. secondary reproductives)

Introduction

Line 41: change "for up to two decades" to e.g. "for several decades". The tables in the Keller 1998 tables indicate that some ant and termite queens manage to live for up to 30 years, and I think there is evidence for >40 in termites.

Line 42: change "admirable lifespan" to either "extraordinary lifespan" or "long lifespan".

Line 47: change "at a time" to "simultaneously".

Line 48: please add some citations for Hymenopteran studies, e.g. from the Foitzik (Temnothorax) and Heinze (Platythyrea, Cardiocondyla) groups.

Line 60: either "the telomerase determines" or "telomerases determine"

Line 63: again, either "The telomerase is" or "Telomerases are"

Line 73: To what extent does "Wnt/ β -catenin pathway" play a role in longevity, anti-oxidation...?

Line 81: remove the sentence "Telomeres and telomerase have rarely been studied in social insects." Which is redundant to the first sentence of this paragraph

Line 83: how about moving the Apis sentence after the cockroach, then you could say "Moreover, we recently..." instead of twice "in our recent study..."

Line 96: add reference to "the recorded maximum was 19 years"

M&M

Line 106 + Line 117: change "are" to "were"

Line 114: how many workers and soldiers?

Line 119: please add coordinates of sampling site. What about sampling permission and Nagoya certificate?

Line 120: change to "were kept in the laboratory"

Line 122: "under a stereomicroscope" and either say "the digestive tube" or "digestive tubes"

Line 126: is "alternative splicings" the correct term? Alternatively "alternative splice variants" or "isoforms"... && add information that you conduct a de novo transcriptome assembly

Line 130: did you run technical replicates?

Line 128 +170+189: "in the electronic"

Line 152: "Pierce BCA assay kit" manufacture information missing

Supplementary Methods

Page2: add information on how DNA was isolated, which protocol did you use, special high molecular weight DNA? Add information on what type of sequencing you are talking about. I assume Nanopore sequencing? How many flow cells?

You extracted DNA from a pool of 8 individuals. Why were “barcoded” libraries loaded, why did you need barcodes for a single library?

“according to 70-day immunization Protocol”: add reference

“by ELISA” + “in Western blot”: add information

Results

Line 195: “mapping on the newly”

Line 195: do you mean transcriptome instead of genome? Otherwise give genome reference

Line 197: does the accession number refer to the psTERT sequence or the transcriptome? Do you make the transcriptome accessible somewhere?

Lines 206-213: The workflow behind these results including the construction of the phylogeny is completely missing from material&methods

Figure 1E = phylogeny: please add the sequence of at least one ant species. Since ants belong to the other eusocial taxa besides termites, it would be highly interesting to them represented in the phylogeny. Please run the ML for at least 1000 replicates.

Line 256: please add the test statistics to the p-value and also add to the M&M section which program and test you used for the statistical analyses.

Discussion

In the beginning of the discussion where you add a short summary of your study add information that you also investigated splice variants, spTERT expression in various tissues and even in multiple species.

Line 286: “of the psTERT protein”

Line 302: “but share exon numbers, junctions and compositions” do you mean “but share the same exon numbers”?

Line 303: “The RBD domain”

Line 314: would it be possible to determine which splice variants are expressed in which tissue, or at least how many of the 6 are expressed in which tissue? If this information could be obtained easily I think it would be an interesting point to address.

Referee: 2

Comments to the Author(s)

The authors studied the potential role of telomere in explaining lifespan differences between queen/king and workers in termites. Social insects are emerging model organisms of ageing as they realize very different lifespan between colony members that share the same genetic background with queens/kings that can have extraordinary longevity for an insect. The authors identified and characterized the telomerase reverse transcriptase (psTERT) in the termite *Prorhinotermes simplex*, quantified psTERT expression, measured telomerase activity and determined telomere lengths and location. They used different caste, tissues and age classes and did some of the experiments also with other termite species. They show that queens and kings have higher telomerase activity, also in most tissues of the soma where it is just as high as in the gonads. But telomere length does not seem to differ between castes.

This is a comprehensive study that addresses a (potentially) important ageing mechanism that received relative low attention in social insects so far and had not been studied in termites.

Therefore, it is well-suited for PRSB. The manuscript is also clear and well-written.

I have a number of questions/topics that should be addressed in a revision, most of them concerning methods and statistics, but also interpretation of results.

Major points

1) I am wondering a bit about the assumption of the study that workers should invest less in anti-ageing. As mentioned in the manuscript, workers in termites like *P. simplex* are immatures from which the reproductives develop (e.g. L. 88). Hence, evolutionary theory predicts that they should invest in anti-aging mechanisms as they are individuals that have not reached maturity yet – in contrast to species with sterile workers or at least a bifurcated development (see also

Monroy Kuhn et al 2019 PLoS One). Please include this aspect into your considerations and conclusions.

2) Does the lifespan between neotenic and primaries really differ? While for the neotenic a range is given for the primaries only a maximum is provided (1.95-97). As you seem to have the data (1.108/109), please add a life table (e.g. Kaplan Meier) analyses. This would also help in interpretation the results (see below)

3) Methods: Many more details are needed in the methods. The samples sizes for each of the experiments should be provided as well as whether the samples came from the same of different (stock)colonies. If samples came from the same colony (or stock colony) this must be included in the analyses as they are not independent. Please provide more information on the genomic sequencing and the assembly protocol as well as the Oxford nanopore methods. Are these data made available? How many colonies did you use for the *Neotermes cubanus*, *Nasutitermes guayanae* and *Prorhinotermes canalifrons*. Did you have to pool sample to analyse, for instance, specific tissues; how many samples did you pool. Please add for each analyses.

4) Statistics: Please add a statistics section to the methods which mentions how the different data sets were analysed. Also add the tests with the typical test characteristics (N or df, F-value...) to the brackets with the P-values in the text. How did you deal with non-independency of data and did you test for assumptions of the tests. Often such data the error distributions of such data are not normal-distributed. Currently the only information provide, is in the legends which is too limited

5) Discussion: (1) One reason why no differences in telomere lengths was found is that the reproductives/workers were not old enough, i.e. no senescence processes were occurring yet. This is a difficult issue as one need survival curves to determine how organisms age. For social insects, there is, e.g. some evidence that suggests that reproductives 'suddenly' age, i.e. there is no gradual senescence. Its, seems that you can do such analyses given that you seem have kept colonies over longs periods. Please, add them to your data. Also mention the ages of the individuals used for the telomere lengths analysis. (2) (partly linked to 1): Another reason why you seen no differences in telomere length between castes may exactly be the telomerase activity. If they protect the telomeres from shortening, then one would not expect to see differences between castes. This might also lead to non-linear scenescene. This is a bit speculative also given that it is currently unclear to me what the sample sizes for the telomere length estimates are. Yet, I think it would be worthwhile considering.

Minor comments

l. 114-115: unclear. Does *P. simplex* has one pair of reproductives or is it polygamous?

l. 117: change to: ...reproductives of known age...

l.145 and the following: Can you please explain how the qTRAP assay functions? How can one do a PCR with proteins?

l. 215-229: Were these the results of single samples? $N = 1$; as implied from Figure 2? Please state clearly. With a $N = 1$ one cannot really say whether it is higher or lower in one caste/ tissue , or not. Can you also show the data for *N. cubanus*.

l. 237: can you specify: how old were they?

Discussion: The discussion is clearly written. I think it would further benefit the reader to have subheadings that characterize the different topics addressed.

l. 300, 301: Please cite the papers which published the two termite genomes. Please, add the original Gene ID as named in the genomes

l. 353: change to: .. phylogenetically basal..

l. 354: The workers of the *Kalotermitidae* are not sterile (see also Abstract, and adjust). In all species with a linear development, they are totipotent immatures. This is important consequences regarding the evolution of ageing; see above and Monroy Kuhn et al 2019 PLoS One.

Author's Response to Decision Letter for (RSPB-2020-2439.R0)

See Appendix A.

RSPB-2021-0511.R0

Review form: Reviewer 1

Recommendation

Accept with minor revision (please list in comments)

Scientific importance: Is the manuscript an original and important contribution to its field?

Excellent

General interest: Is the paper of sufficient general interest?

Good

Quality of the paper: Is the overall quality of the paper suitable?

Excellent

Is the length of the paper justified?

Yes

Should the paper be seen by a specialist statistical reviewer?

No

Do you have any concerns about statistical analyses in this paper? If so, please specify them explicitly in your report.

No

It is a condition of publication that authors make their supporting data, code and materials available - either as supplementary material or hosted in an external repository. Please rate, if applicable, the supporting data on the following criteria.

Is it accessible?

Yes

Is it clear?

Yes

Is it adequate?

Yes

Do you have any ethical concerns with this paper?

No

Comments to the Author

The authors did a great job in revising the manuscript.

I only have three minor comments concerning the discussion

I) Add references to both sentences

333 strong nuclear localization signal RRRKKKIK in the N-terminal region of TSS2 variants suggests the
 334 transport of the protein through nuclear pores, while the signal peptide
 335 MFRSCLTIFRVRCYRAVVFVWVTLCSLLTDC in TSS1 isoforms may be linked to vesicular
 transport to
 336 other membrane-bound organelles. In conclusion, psTERT structure indicates the ability of
 psTERT to
 337 perform its canonical function, but also some features associated with non-canonical
 functions
 338 remained conserved.

II) check grammar

It appears,

380 thus, that the rate of telomerase somatic upregulation in reproductives is correlated with
 lifespan

381 differences between working and reproductive castes, fertility and longevity of the
 reproductives." => the last part of the sentence does not seem to be correct

III) line 387: change "maxima" to "maximum"

Review form: Reviewer 2 (Judith Korb)

Recommendation

Accept as is

Scientific importance: Is the manuscript an original and important contribution to its field?

Excellent

General interest: Is the paper of sufficient general interest?

Excellent

Quality of the paper: Is the overall quality of the paper suitable?

Excellent

Is the length of the paper justified?

Yes

Should the paper be seen by a specialist statistical reviewer?

No

Do you have any concerns about statistical analyses in this paper? If so, please specify them explicitly in your report.

No

It is a condition of publication that authors make their supporting data, code and materials available - either as supplementary material or hosted in an external repository. Please rate, if applicable, the supporting data on the following criteria.

Is it accessible?

Yes

Is it clear?

Yes

Is it adequate?

Yes

Do you have any ethical concerns with this paper?

No

Comments to the Author

Many thanks for the responses and careful revision that addressed all my comments. I am looking forward to see this study published in Proceedings of the Royal Society B.

Decision letter (RSPB-2021-0511.R0)

25-Mar-2021

Dear Dr Hanus

I am pleased to inform you that your manuscript RSPB-2021-0511 entitled "Long-lived termite kings and queens activate telomerase in somatic organs" has been accepted for publication in Proceedings B.

The referees have recommended publication, but one referee has requested some minor revisions to your manuscript. Therefore, please respond to the referee's comments and revise your manuscript. Because the schedule for publication is very tight, it is a condition of publication that you submit the revised version of your manuscript within 7 days. If you do not think you will be able to meet this date please let us know.

- 1) A text file of the manuscript (doc, txt, rtf or tex), including the references, tables (including captions) and figure captions. Please remove any tracked changes from the text before submission. PDF files are not an accepted format for the "Main Document".
- 2) A separate electronic file of each figure (tiff, EPS or print-quality PDF preferred). The format should be produced directly from original creation package, or original software format. PowerPoint files are not accepted.
- 3) Electronic supplementary material: this should be contained in a separate file and where possible, all ESM should be combined into a single file. All supplementary materials accompanying an accepted article will be treated as in their final form. They will be published alongside the paper on the journal website and posted on the online figshare repository. Files on

figshare will be made available approximately one week before the accompanying article so that the supplementary material can be attributed a unique DOI.

Yours sincerely,

Professor Loeske Kruuk

Editor

Associate Editor

Board Member

Comments to Author:

I want to thank the authors for careful consideration of reviewers' comments and the reviewers for their outstanding efforts. I am happy with the current version of this paper and do not have any further comments. Please pay attention to the minor revisions suggested by Reviewer 1.

Reviewer(s)' Comments to Author:

Referee: 1

Comments to the Author(s).

The authors did a great job in revising the manuscript.

I only have three minor comments concerning the discussion

I) Add references to both sentences

333 strong nuclear localization signal RRRKKKIK in the N-terminal region of TSS2 variants suggests the

334 transport of the protein through nuclear pores, while the signal peptide

335 MFRSCLTIFRVRCYRAVVFVVTLCSLLTDC in TSS1 isoforms may be linked to vesicular transport to

336 other membrane-bound organelles. In conclusion, psTERT structure indicates the ability of psTERT to

337 perform its canonical function, but also some features associated with non-canonical functions

338 remained conserved.

II) check grammar

It appears,

380 thus, that the rate of telomerase somatic upregulation in reproductives is correlated with lifespan

381 differences between working and reproductive castes, fertility and longevity of the reproductives." => the last part of the sentence does not seem to be correct

III) line 387: change "maxima" to "maximum"

Referee: 2

Comments to the Author(s).

Many thanks for the responses and careful revision that addressed all my comments. I am looking forward to see this study published in Proceedings of the Royal Society B.

Author's Response to Decision Letter for (RSPB-2021-0511.R0)

See Appendix B.

Decision letter (RSPB-2021-0511.R1)

26-Mar-2021

Dear Dr Hanus

I am pleased to inform you that your manuscript entitled "Long-lived termite kings and queens activate telomerase in somatic organs" has been accepted for publication in Proceedings B.

You can expect to receive a proof of your article from our Production office in due course, please check your spam filter if you do not receive it. PLEASE NOTE: you will be given the exact page

length of your paper which may be different from the estimation from Editorial and you may be asked to reduce your paper if it goes over the 10 page limit.

Data Accessibility section

Open Access

Paper charges

Sincerely,

Appendix A

RSPB-2020-2439

Response to reviewer's comments

Reviewer(s)' Comments to Author:

Referee: 1

Comments to the Author(s)

Major comments

1. You make use of a multitude of different methods, and you make use of multiple species which is great. Emphasize this at the end of the introduction and the beginning of the discussion. At the moment the connection and reasoning of the different approaches is not always clear.

RESPONSE (1): Thank you for this suggestion. We extended a little bit the closing paragraph of the Introduction so as to underline the complementary contribution of the various methods to the goals of the study and to put forward that we try to cover as much as possible the developmental dynamics leading to different castes in the complex caste system of our model, including the focus on individual somatic/germ organs. We also highlight the two additional species and their different biology, which allows us making more general conclusions/comparisons.

Likewise, we emphasized the different approaches and multiple species used at the beginning of the Discussion (see also response 35).

2. Some information in the material and methods section is missing including the statistical analyses (only rudimentary mentioned in the text, some details only in the figure legend).

RESPONSE (2): We admit that our attempt to save the text space and restrict the statistical procedures to a minimum in the figure legends was not appropriate. The revised version of the manuscript now contains: (1) a list of sample sets used for all different techniques (qTRAP, ELISA, qPCR), in which stats were utilized to interpret the results (the second chapter in the electronic supplementary material, Methods), along with the description of the tests used, data transformation, test assumption verifications and software, (2) in the text body of the results, the most important comparisons are supported with appropriate test statistics (ANOVA F-value with n and DF, Dunnett's test q-values with n, p-values), and (3) supplementary tables S4 and S5 provide a list of all test statistics calculated and are referred to in the results section and figure legends. Likewise, the other details on methods requested by the reviewer in individual points below were completed.

3. Information on the RNAseq data and transcriptome are not complete including data deposition information.

RESPONSE (3): We agree that this aspect needed clarifications/completions, which are provided in responses to individual minor comments below.

Here, we also summarize the two sequencing projects used in this study, specify where the related information can be found in the original and revised versions of the manuscript and which parts of the data will be made publicly available upon eventual publication of our study. In this study, we used:

A) Draft genome assembly obtained from Oxford Nanopore sequencing of pooled worker samples. Methods of the sequencing and assembly are specified in the electronic supplementary information. The complete *P. simplex* genome annotation is currently underway and will be (hopefully) published separately. Yet, we take the liberty to mine (and publish) individual genes for the purposes of running projects, as we did previously, e.g., with

P. simplex insulin receptor genes (Smykal et al. 2020, DOI: 10.1093/molbev/msaa048). We now make clear in the text that we were working with an unpublished draft genome assembly, from which we reconstructed the psTERT gene structure, available under the provided GenBank accession number (this will be made publicly accessible upon eventual publication of the manuscript). The GenBank entry also specifies all the six identified transcript variants.

B) Illumina-generated transcriptomes of queen ovaries and worker abdominal soma to reconstruct the psTERT gene structure and identify transcript variants and putative psTERT protein isoforms. The RNA Seq data used are a part of a larger project on quantitative comparison of transcriptomes from workers, kings and queens of different ages (heads, abdominal soma, gonads), used in a study on caste/age related transcription dynamics, which will be published separately. Nevertheless, since we used here the queen ovaries and worker soma RNA Seq data, we now make accessible the reads from these two phenotypes as SRA archives. Their accession number is now stated in the Data accessibility section of the main text, and will be made publicly accessible upon eventual publication of the manuscript. Assembly techniques were already described in the electronic supplementary file under “RNA and DNA isolations” and “Next-generation sequencing of RNA and DNA, data analysis” sections.

Minor comments:

Abstract

First sentence: maybe rather start with eusocial insects in general and not directly with termites in particular

RESPONSE (4): Yes, it is a relevant suggestion. The broader comparison with other advanced eusocial insects is desirable (just as we do so in the introduction), yet not easy due to the space limitation of the abstract. We revised the first sentence of the abstract to put forward the analogy between termite reproductives and queens of advanced eusocial Hymenoptera.

Line 28: rephrase “neotenic” reproductives for non-termite experts (i.e. secondary reproductives)

RESPONSE (5): We agree. We replaced “neotenic” for “secondary”. The reader then learns on the ontogenetic origin of “secondary” reproductives and the synonymy with “neotenic” in the introduction.

Introduction

Line 41: change “for up to two decades” to e.g. “for several decades”. The tables in the Keller 1998 tables indicate that some ant and termite queens manage to live for up to 30 years, and I think there is evidence for >40 in termites.

RESPONSE (6): We changed “for up to two decades or more”. We are aware of the content of the Keller 1998 review but we prefer the more conservative of the observations reported therein, since for the one estimating 30 years for the termite queen (and some other claims on 40 to 50 years lifespan) there are some doubts on the source of information. We thus prefer the data made on systematically surveyed laboratory colonies. Also our own observations reported here suggest a survival of reproductives for roughly two decades in our model species.

Line 42: change “admirable lifespan” to either “extraordinary lifespan” or “long lifespan”.

RESPONSE (7): We changed to “extraordinary”.

Line 47: change “at a time” to “simultaneously.”

RESPONSE (8): We exchanged “at a time” for “accompanied by”...

Line 48: please add some citations for Hymenopteran studies, e.g. from the Foitzik (Temnothorax) and Heinze (Platythyrea, Cardiocondyla) groups.

RESPONSE (9): We added three references (out of many possible candidates) on original papers or reviews on ants (including *Temnothorax* and *Cardiocondyla*) or bees.

Line 60: either “the telomerase determines“ or “telomerases determine”

RESPONSE (10): We added “the”...

Line 63: again, either “The telomerase is” or “Telomerases are”

RESPONSE (11): We added “The”...

Line 73: To what extent does “Wnt/ β -catenin pathway” play a role in longevity, anti-oxidation...?

RESPONSE (12): We did not mean that Wnt/ β -catenin pathway plays any specific role in longevity or anti-oxidation, we just wanted to list the non-canonical functions ascribed in the literature to the telomerase. To make this clear, we modified “non-canonical pro-longevity functions” to “non-canonical pro-longevity and regulatory functions”.

Line 81: remove the sentence “Telomeres and telomerase have rarely been studied in social insects.” Which is redundant to the first sentence of this paragraph

RESPONSE (13): The sentence is now removed and the paragraph restructured.

Line 83: how about moving the *Apis* sentence after the cockroach, then you could say “Moreover, we recently...” instead of twice “in our recent study...”

RESPONSE (14): The paragraph is now restructured accordingly.

Line 96: add reference to “the recorded maximum was 19 years”

RESPONSE (15): It is our own observation that we report here for the first time. Since the paragraph has been reworded on request of Reviewer 2, this information should now be apparent.

M&M

Line 106 + Line 117: change “are” to “were”

RESPONSE (16): Corrected.

Line 114: how many workers and soldiers?

RESPONSE (17): Information completed (Thirty workers plus five soldiers). This text is now a part of the first section of supplementary information.

Line 119: please add coordinates of sampling site. What about sampling permission and Nagoya certificate?

RESPONSE (18): The area of sampling is now specified including the range of GPS coordinates. The field work in F. Guiana took place outside the Guiana Amazonian Park and was realized with the regularly renewed consent of Office National des Fôrets (Cayenne), allowing the field work to R. Hanus and J. Křivánek, and treating, among others, the use of

the acquired material. This is now stated in the Methods section as a part of the first section of supplementary information.

Line 120: change to “were kept in the laboratory”

RESPONSE (19): Corrected. This text is now a part of the first section of supplementary information.

Line 122: “under a stereomicroscope” and either say “the digestive tube” or “digestive tubes”

RESPONSE (20): Corrected. This text is now a part of the first section of supplementary information.

Line 126: is “alternative splicings” the correct term? Alternatively “alternative splice variants” or “isoforms”... && add information that you conduct a de novo transcriptome assembly

RESPONSE (21): The sentence is now reworded accordingly.

Line 130: did you run technical replicates?

RESPONSE (22): Yes, qPCRs were performed in technical duplicates, this information is now completed.

Line 128 +170+189: “in the electronic”

RESPONSE (23): Corrected

Line 152: “Pierce BCA assay kit” manufacture information missing

RESPONSE (24): Completed.

Supplementary Methods

Page2: add information on how DNA was isolated, which protocol did you use, special high molecular weight DNA? Add information on what type of sequencing you are talking about. I assume Nanopore sequencing? How many flow cells?

RESPONSE (25): In fact, the requested information is already provided in the original version of the Supplementary information, page 2 (now pages 5 and 6). Nevertheless, the first sentence in the second paragraph (DNA isolations) is now reworded to make clear that the high molecular DNA was destined for Oxford Nanopore genomic sequencing. Details on this sequencing (including flow cell number) were also in the original version of the supplementary file, page 2 and 3 (now pages 5 and 6).

You extracted DNA from a pool of 8 individuals. Why were “barcoded” libraries loaded, why did you need barcodes for a single library?

RESPONSE (26): We agree that the sentence was unclear, it is now reworded to make clear that we generated two libraries, one native (very long reads, lower yields) and one PCR amplified (shorter reads, higher yields), which were subsequently pooled prior to sequencing.

“according to 70-day immunizationProtocol“: add reference

RESPONSE (27): Instead of referring to the supplier’s proprietary protocol we now provide technical details on the immunization scheme used for both antibodies.

“by ELISA” + “in Western blot”: add information

RESPONSE (28): Specific details for both techniques are added.

Results

Line 195: “mapping on the newly”

RESPONSE (29): The sentence is now reworded.

Line 195: do you mean transcriptome instead of genome? Otherwise give genome reference

RESPONSE (30): RNASeq and RACE data were mapped on our in-house draft genome assembly. As stated above (3), we now make clear in this sentence that we were working with an unpublished draft genome assembly, from which we reconstructed the psTERT gene structure, available under the provided GenBank accession number, along with 6 transcript variants.

Line 197: does the accession number refer to the psTERT sequence or the transcriptome? Do you make the transcriptome accessible somewhere?

RESPONSE (31): It should now be clear that the accession number refers to the psTERT gene. As explained above, the RNA Seq data used for psTERT reconstruction are part of a larger transcriptomic project. The queen and worker reads used here are made accessible as SRA archives, referred to in the methods and in data availability section of the main text.

Lines 206-213: The workflow behind these results including the construction of the phylogeny is completely missing from material&methods

RESPONSE (32): We added a few sentences on protein sequence alignment and exploration as well as on the construction of the phylogenetic tree in the Supplementary Methods (Next-generation sequencing of RNA and DNA, data analysis). We also added a new table S3, which shows the accession numbers for TERT protein sequences used in the phylogeny.

Figure 1E = phylogeny: please add the sequence of at least one ant species. Since ants belong to the other eusocial taxa besides termites, it would be highly interesting to them represented in the phylogeny. Please run the ML for at least 1000 replicates.

RESPONSE (33): It is true that ants should be represented in the phylogeny. Therefore, we added *Monomorium pharaonis* TERT protein sequence in the alignment and resulting phylogenetic tree. The ML calculations are now based on 1000 replicates.

Line 256: please add the test statistics to the p-value and also add to the M&M section which program and test you used for the statistical analyses.

RESPONSE (34): As stated above, we now provide the test statistics for the most important findings in the text and for analyses in the supplementary tables S4 and S5, and the statistics are now described in a separate chapter, specifying the sample sets, data transformations and treatment, software, etc.

Discussion

In the beginning of the discussion where you add a short summary of your study add information that you also investigated splice variants, spTERT expression in various tissues and even in multiple species.

RESPONSE (35): We added this information at the beginning of the discussion.

Line 286: “of the psTERT protein”

RESPONSE (36): Corrected.

Line 302: “but share exon numbers, junctions and compositions” do you mean “but share the same exon numbers”?

RESPONSE (37): Corrected

Line 303: “The RBD domain”

RESPONSE (38): Corrected.

Line 314: would it be possible to determine which splice variants are expressed in which tissue, or at least how many of the 6 are expressed in which tissue? If this information could be obtained easily I think it would be an interesting point to address.

RESPONSE (39): This is a relevant suggestion. In fact, the differential expression of the detected splice variants of psTERT (eventually presence of undetected ones) is one of the topics of the doctoral project of the first co-author of the present manuscript (M. Pangrácová) and it is currently being studied. We possess the initial data from the RNA Seq analysed using DEXseq algorithm, which suggest differential exon usage of psTERT in somatic vs. reproductive organs. The running research thus builds on this promising observation in an attempt to verify experimentally the expression differences among phenotypes/tissues on the one hand and a potential differential cellular localization of psTERT protein isoforms on the other. The outputs should make up a new future publication.

We now re-inserted the sentence “differential expression of the multiple identified splice variants leading to several potential TERT isoforms with different cell localization signals and possibly also functions should be studied in individual life stages and organs and with respect to life expectancy and actual age of the termites” in the closing paragraph (perspectives) of the discussion, a sentence which has been included in the draft versions but did not survive the shortening to meet the page limits for PRSB.

Referee: 2

Comments to the Author(s)

The authors studied the potential role of telomere in explaining lifespan differences between queen/king and workers in termites. Social insects are emerging model organisms of ageing as they realize very different lifespan between colony members that share the same genetic background with queens/kings that can have extraordinary longevity for an insect. The authors identified and characterized the telomerase reverse transcriptase (psTERT) in the termite *Proterhinotermes simplex*, quantified psTERT expression, measured telomerase activity and determined telomere lengths and location. They used different caste, tissues and age classes and did some of the experiments also with other termite species. They show that queens and kings have higher telomerase activity, also in most tissues of the soma where it is just as high as in the gonads. But telomere length does not seem to differ between castes.

This is a comprehensive study that addresses a (potentially) important ageing mechanism that received relative low attention in social insects so far and had not been studied in termites. Therefore, it is well-suited for PRSB. The manuscript is also clear and well-written. I have a number of questions/topics that should be addressed in a revision, most of them concerning methods and statistics, but also interpretation of results.

Major points

1) I am wondering a bit about the assumption of the study that workers should invest less in anti-ageing. As mentioned in the manuscript, workers in termites like *P. simplex* are immatures from which the reproductives develop (e.g. L. 88). Hence, evolutionary theory predicts that they should invest in anti-ageing mechanisms as they are individuals that have not reached maturity yet – in contrast to species with sterile workers or at least a bifurcated development (see also Monroy Kuhn et al 2019 PLoS One). Please include this aspect into your considerations and conclusions.

RESPONSE (40): Thank you for this relevant comment. When writing the original version of our manuscript, we were of course aware of the paper by Monroy Kuhn et al. (2019), and the observations reported therein. In the draft version of our manuscript we discussed our findings in the light of the mentioned paper, yet this discussion did not survive the shortening of the text so as to meet the page limit of PRSB. In the revised version, we reformulate our starting hypotheses in the context of this paper by adding a paragraph by the end of the Introduction section.

In this new paragraph, we now appropriately cite the Monroy Kuhn et al. (2019) paper and its conclusions that the “pseudergate” working immatures do display, in agreement with the theories, upregulation in some repair and pro-longevity functions, which may allow them a future long lifespan upon the potential development into reproductives (primary of secondary). Our starting hypothesis (and conclusions) *à priori* do not stand in contrast. Our assumption stated in this new paragraph is that the readiness for long life expected in immature working phenotypes does not preclude the recruitment of further maintenance/repair mechanisms during the maturation and maturity of the reproductives.

In fact, it is reasonable to expect that immatures will invest into the somatic maintenance (for future reproduction) proportionally to their chances to become reproductive, just as they do so in terms of their readiness to reproduce = development of gonads, etc. And that additional somatic maintenance/repair mechanisms will be upregulated once the reproductive option becomes real and they differentiate into a reproductive (again, just as they do so in terms of their readiness to reproduce = development of gonads, etc...). Therefore, we also state that it

is the maturation/maturity stage of the reproductives which is the target of our interest, rather than later stages of their life when more signs of senescence are to be expected.

We also added to the discussion a paragraph, which considers our findings on telomerase activation in context of the investment of immatures into pro-longevity mechanisms. We state there that interspecific comparison of somatic telomerase activation in reproductives of three phylogenetically distant species is in fact in line with the reasoning and observations reported in Monroy Kuhn et al. (2019) paper: in species with high reproductive chances of working immatures (such as Kalotermitidae) the relative level of somatic activation of telomerase in reproductives is the lowest, followed by species with lower reproductive potential of immatures (Rhinotermitidae: *Prorhinotermes*) and with no reproductive potential (workers in Termitidae).

2) Does the lifespan between neotenic and primaries really differ? While for the neotenic a range is given for the primaries only a maximum is provided (1.95-97). As you seem to have the data (1.108/109), please add a life table (e.g. Kaplan Meier) analyses. This would also help in interpretation the results (see below)

RESPONSE (41): We agree that the wording of the original sentence was unfortunate and may lead to the conclusion that the lifespans of neotenic and primaries are substantially different. In fact, we cannot claim this from our available data. And we also agree that a more tangible presentation of the life expectancies of reproductives may be beneficial. Therefore, in the revised manuscript we include survival curves of reproductives for 15 colonies headed by primaries. Most of these colonies were kept intact and were disturbed only to control the survival (or eventual replacement by neotenic) of the kings and queens during the past 20 years. Only from a few of them the primaries were removed for scientific purposes and are treated as censored objects in the survival analysis. This allows us to estimate the median survival for the primary king and queen. By contrast, the situation is more complicated with the neotenic reproductives because large proportion of neotenic of known age (starting by date of replacement primary king or queen by same-sex neotenic) has been “censored” for various techniques used in this project (and other projects) before reaching the age of 10 years. Therefore, we have solid observations that they practically always survive 5 years and may live up to 10 years, but we do not know the genuine maxima of their lifespan which would allow the construction of survival curves and rigorous comparison with primaries. In the revised version of the introduction, we reworded the survival section (Introduction) accordingly, we give the median survival for the primary king and queen, together with the actual maximum, and we provide the reference to the supplementary figure S1, which shows the survival curve for primaries.

As we also state above, we focus our study on the somatic maintenance mechanisms recruited during the maturation and the maturity of the reproductives, which would allow them to extend their lifespan. Therefore, for our analyses we were selecting reproductives from the maturity period, usually between 3 and 5 years after differentiation (eventually younger when we were interested in the dynamics during the maturation), which thus was long before the potential onset of senescence. This range of ages is now given in the Introduction as well in the supplementary chapter “Analysed sample sets and statistical evaluation”, specific age is given in the figure legends.

3) *Methods: Many more details are needed in the methods.* The samples sizes for each of the experiments should be provided as well as whether the samples came from the same or different (stock)colonies. If samples came from the same colony (or stock colony) this must be included in the analyses as they are not independent. Please provide more information on the genomic sequencing and the assembly protocol as well as the Oxford nanopore methods.

Are these data made available? How many colonies did you use for the *Neotermes cubanus*, *Nasutitermes guayanae* and *Prorhinotermes canalifrons*. Did you have to pool sample to analyse, for instance, specific tissues; how many samples did you pool. Please add for each analyses.

RESPONSE (42): We agree that the sampling design (number of colonies, number of individuals sampled) was not clear for all analyses in the original submission, and in attempt to save space we included some information only in the legends of figures without appropriately stating them in the text itself. Because we used multiple different techniques, we decided in this revision to review the sampling design, number of repetitions, ages of sampled individuals, etc., on one place. Thus, we created a new section in the supplementary information (Analysed sample sets and statistical evaluation), which provides the number of colonies used to obtain the data, the stages and castes used, their number per one sample (if pooling of material was needed), number of replicates, etc. This section states that for overall qTRAP and TRF comparison among castes we used multiple colonies of the given species, while detailed TA in different organs, psTERT expression, psTERT abundance and TA during the development was always studied on multiple individuals from one colony.

This section also describes the strategy how we designed the comparisons of different colonies, i.e. that we focused all analyses on the relative differences between the observations in different castes and life stages and the mean values obtained for workers from the respective colony. The workers thus serve as a “control phenotype” in all analyses and as a “scaling factor“ among colonies and also among different multiwell plates prepared from the same colony. This is reflected also in the statistical treatment of the resulting data, summarized in the same section: the data for different phenotypes, usually after log₂ transformation, is compared to the worker phenotype using Dunnett’s multiple comparison test.

The original version of the Online Supplementary Material contains under “Next-generation sequencing of RNA and DNA, data analysis” the basic information on gDNA isolation, library preparations, Oxford Nanopore sequencing, raw sequence treatment, demultiplexing, and assembly. We added to this paragraph, on request of Reviewer 1, a clarification on the preparation of the two different types of libraries.

4) Statistics: Please add a statistics section to the methods which mentions how the different data sets were analysed. Also add the tests with the typical test characteristics (N or df, F-value...) to the brackets with the P-values in the text. How did you deal with non-independency of data and did you test for assumptions of the tests. Often such data the error distributions of such data are not normal-distributed. Currently the only information provide, is in the legends which is too limited.

RESPONSE (43): We admit that our attempt to save the text space and restrict the statistical procedures to a minimum in the figure legends was not appropriate. The revised version of the manuscript now contains:

(1) a list of sample sets used for all different techniques (qTRAP, ELISA, qPCR), in which stats were utilized to interpret the results (a separate chapter „Analysed sample sets and statistical evaluation“ in the electronic supplementary material), along with the description of the tests used, data transformation, test assumption verifications, software, strategy how to relate datasets from different colonies by expressing the relative changes when compared to worker phenotype

(2) in the text body of the results, the most important comparisons are supported with appropriate test statistics (ANOVA F-value with n and DF, Dunnett’s test q values with n, p-values)

(3) supplementary tables S4 and S5 provide a list of all test statistics calculated and are referred to in the results section and figure legends.

As stated in the newly included „Analysed sample sets and statistical evaluation“ chapter, all datasets were checked for the basic assumptions for parametric analyses of variance, i.e. equal variances (Brown-Forsythe test) and distribution normality (Wilk-Shapiro test) prior and after data transformation (log₂ transformation to correct for heteroscedasticity).

5) Discussion: (1) One reason why no differences in telomere lengths was found is that the reproductives/workers were not old enough, i.e. no senescence processes were occurring yet. This is a difficult issue as one needs survival curves to determine how organisms age. For social insects, there is, e.g. some evidence that suggests that reproductives ‘suddenly’ age, i.e. there is no gradual senescence. It seems that you can do such analyses given that you seem to have kept colonies over long periods. Please, add them to your data. Also mention the ages of the individuals used for the telomere lengths analysis.

(2) (partly linked to 1): Another reason why you see no differences in telomere length between castes may exactly be the telomerase activity. If they protect the telomeres from shortening, then one would not expect to see differences between castes. This might also lead to non-linear senescence. This is a bit speculative also given that it is currently unclear to me what the sample sizes for the telomere length estimates are. Yet, I think it would be worthwhile considering.

RESPONSE (44) to points (1) and (2): As we state elsewhere, our focus in this study was to consider the differences between the working stages and the reproductives during their maturation and maturity to see what are the (potential) pro-longevity mechanisms recruited during these stages to see what they do to extend their lifetime by several more years. Therefore, even though we have an idea on the longevity of the reproductives, which can easily reach over a decade, we selected fully mature individuals (in terms of reproduction output) of ages mostly 3-5 years, rather than senescent reproductives. Moreover, a parallel focus on senescent reproductives and processes would also be beyond our capacities in terms of the available material.

Nevertheless, the main message of our observations on telomere sizes, which we state in the discussion, is that the telomeres in termites really are relatively long regardless of the caste and age of the considered individuals, when compared for instance with humans having 5-15kb. This raises doubts whether gradual telomere shortening may even be a life-limiting factor in termites. As we state in the discussion, such relatively long telomeres can also be observed in other advanced Polyneoptera (mantises, stick insects, cockroaches). Therefore, we speculate in the discussion that (at least) the whole crown clade of Polyneoptera may be somewhat preadapted for relatively long lifespan, and the telomeres are not the active regulators of lifespan at all.

In fact, to the best of our knowledge, there is no evidence on the causal link between telomere shortening and lifespan in insects as a whole. This is related to the second main message, i.e. that the considerable length of termite telomeres applies to all phenotypes, independently of the level of telomerase activation and that the telomerase is activated even in tissues with low nuclear division frequencies in adults, where it is quite unexpected, just as it is unexpected in adult soma in general. This leads us to the conclusion that the telomerase activation observed in termite reproductives and honeybee queens may be independent of its canonical role.

Nevertheless, to provide at least some answer to the raised question, we now added to the manuscript a new panel in figure 2 (Fig. 2D), which compares the telomere sizes, estimated by means of TRF, in *P. simplex* workers (pseudergates) of two different stages: a young stage 4, one of the most abundant stages encountered in the colonies and the stage 7, one of the two oldest stages regularly found in the colonies. This figure shows that also in the relatively old

workers of stage 7 there is no important shortening of telomeres which might reach the values known to lead to eventual replicative senescence (units of kb). We added the same analysis also for the other termite species examined, i.e. *Prorhinotermes canalifrons* and *Neotermes cubanus*, as a separate panel in figure S4C, showing once again analogous results to those for *P. simplex*.

With respect to ages and sample sizes: We regret having omitted in the original submission the information on sampling for the three studied species and on the design of the TRF gels/membranes. This is now corrected by adding following information: (a) the separate chapter „Analysed sample sets and statistical evaluation/Telomere lengths estimation” in the supplementary information specifies the number of colonies examined (7 for *P. simplex*, 5 for *P. canalifrons*, and 5 for *N. cubanus*). In fact, for *P. simplex*, many more colonies were partially sampled and analysed during the initially stages of the research and fine-tuning of the method. This chapter also specifies which castes were sampled from each species, how many individuals were pooled for gDNA extraction, etc., and describes the design of the gels (what types of samples were compared in two different gels in each colony), (b) Ages of the reproductives are given also in figure legends (fig. 2 and supplementary figs. S4), (c) Ages of reproductives studied are mentioned in the Introduction section in the newly added closing paragraph of the Introduction.

Minor comments

l. 114-115: unclear. Does *P. simplex* has one pair of reproductives or is it polygamous?

RESPONSE (45): Here, we do not understand the question. In *P. simplex*, the pair of primaries is usually replaced by multiple male and female neotenics (up to more than 10 of each sex in large colonies). Therefore, also during the “social induction” of neotenics in orphaned groups of workers, several neotenics of both sexes start to differentiate after ca 10 days. Even though multiple mature neotenics are normally tolerated in the colony, it does not apply to the initial period of their maturation, during which the neotenics perform siblicidal fights. To prevent this, freshly moulted neotenics are placed as two pairs (a number which usually does not trigger the fights) into new groups of helpers, which are freshly removed from original colonies and regularly exchanged so as prevent their differentiation into further neotenics. To make this more clear, we slightly modified the text, which now states that “MULTIPLE neotenics started to differentiate...” and that “Newly moulted neotenics were removed every 24 hours and introduced AS TWO PAIRS into culture groups”. These details of breeding and social induction are now a part of the first section of the supplementary information.

l. 117: change to: ...reproductives of known age...

RESPONSE (46): Modified.

l.145 and the following: Can you please explain how the qTRAP assay functions? How can one do a PCR with proteins?

RESPONSE (47): qTRAP is a relatively established technique, which replaced the traditional TRAP assay. It measures the activity of the enzyme by quantifying its catalytic product, i.e. the TTAGG_n (or TTAGGG_n in vertebrates) sequence added by telomerase to the forward telomerase substrate (TS) primer when incubated with a pool of nucleotides. The elongation product is quantified by RT qPCR using specific primers. The method takes advantage of the amplification steps and thus detects even very low activities of the enzyme and allows using single non-pooled samples even for small organs such as testes.

The Methods section provides a short description including reference to our recent paper. We now modified this text to make it more intelligible by splitting it into two sentences, one

speaking about the incubation and the other one about the amplification and quantification of the elongation product. The text is now a part of the Supplementary methods.

l. 215-229: Were these the results of single samples? $N = 1$; as implied from Figure 2? Please state clearly. With a $N = 1$ one cannot really say whether it is higher or lower in one caste/tissue, or not. Can you also show the data for *N. cubanus*.

RESPONSE (48): Please, see first the response (44) above for the number of colonies used for TRF and the discussion on telomere lengths in general. As we state in the supplementary chapter „Analysed sample sets and statistical evaluation/Telomere lengths estimation”, we designed from each studied colony two different electrophoretic gels and resulting Southern hybridization membranes. One was designed to compare the soma (abdomens or abdomens and heads separately) of all sampled castes from the colony and gonads of reproductives. Even though the samples are usually present only as singletons on these membranes (due to limited capacity of wells on gels), similar results were observed for multiple membranes prepared with samples originating from different colonies. The other was comparing two to three independent replicates of workers of two different stages, i.e. stage 4 and stage 7. ...

As for the data on *N. cubanus*: As we now state in supplementary chapter „Analysed sample sets and statistical evaluation/Telomere lengths estimation”, five colonies were examined. Yet, the resulting Southern hybridization membranes were not of publishable quality. Therefore, we sampled two more colonies and prepared new gels and membranes during this revision. The results are now presented as two figures, the first one (Fig. S4B) shows the comparison of different castes and the second one (Fig. 4C) compares three independent samples of workers of stages 4 and 7.

l. 237: can you specify: how old were they?

RESPONSE (49): Their age was between three and four years, so the compromise labelling would be three-years-old neotenics, which is now stated in the text.

Discussion: The discussion is clearly written. I think it would further benefit the reader to have subheadings that characterize the different topics addressed.

RESPONSE (50): We now structured the Discussion section into four subchapters with separate subheadings.

l. 300, 301: Please cite the papers which published the two termite genomes. Please, add the original Gene ID as named in the genomes

RESPONSE (51): Completed.

l. 353: change to: .. phylogenetically basal..

RESPONSE (52): Changed.

l. 354: The workers of the Kalotermitidae are not sterile (see also Abstract, and adjust). In all species with a linear development, they are totipotent immatures. This is important consequences regarding the evolution of ageing; see above and Monroy Kuhn et al 2019 PLoS One.

RESPONSE (53): We understand. Of course, we are aware of the linear developmental pattern in Kalotermitidae and the developmental potential of its immatures. In fact, when writing about termites, it is sometimes difficult to use appropriate terminology without making the text difficult to understand to non-specialists (e.g. neotenics, pseudergates, etc.) and vice versa, not to oversimplify the complex termite caste systems when using the common terminology. We now reviewed the text (including Abstract) and tried to replace the

terms “workers“ and “sterile” where their potential future reproduction should be apparent by “working immatures”, “immatures”, “working stages”, etc... In the case mentioned here, we used the term “Working immatures”.

Appendix B

RSPB-2021-0511

Reviewer(s)' Comments to Author:

Referee: 1

Comments to the Author(s).

The authors did a great job in revising the manuscript.

I only have three minor comments concerning the discussion

I) Add references to both sentences

333 strong nuclear localization signal RRRKKKIK in the N-terminal region of TSS2 variants suggests the
334 transport of the protein through nuclear pores, while the signal peptide

335 MFRSCLTIFRVRCYRAVVFVVTLCSSLTDC in TSS1 isoforms may be linked to vesicular
transport to

336 other membrane-bound organelles. In conclusion, psTERT structure indicates the ability of psTERT to
337 perform its canonical function, but also some features associated with non-canonical functions

338 remained conserved.

RESPONSE: We added the requested references.

II) check grammar

It appears,

380 thus, that the rate of telomerase somatic upregulation in reproductives is correlated with lifespan

381 differences between working and reproductive castes, fertility and longevity of the reproductives." =>

the last part of the sentence does not seem to be correct

RESPONSE: The sentence has been reworded.

III) line 387: change "maxima" to "maximum"

RESPONSE: Corrected.

Referee: 2

Comments to the Author(s).

Many thanks for the responses and careful revision that addressed all my comments. I am looking forward to see this study published in Proceedings of the Royal Society B.